# Research on Ground Object Classification Method of High Resolution Remote-Sensing Images Based on Improved DeeplabV3+

**DOI:** 10.3390/s22197477

**Published:** 2022-10-02

**Authors:** Junjie Fu, Xiaomei Yi, Guoying Wang, Lufeng Mo, Peng Wu, Kasanda Ernest Kapula

**Affiliations:** 1College of Mathematics and Computer Science, Zhejiang A and F University, Hangzhou 311300, China; 2Key Laboratory of Forestry Intelligent Monitoring and Information Technology of Zhejiang Province, Hangzhou 311300, China

**Keywords:** high-resolution remote-sensing images, semantic segmentation, object classification

## Abstract

Ground-object classification using remote-sensing images of high resolution is widely used in land planning, ecological monitoring, and resource protection. Traditional image segmentation technology has poor effect on complex scenes in high-resolution remote-sensing images. In the field of deep learning, some deep neural networks are being applied to high-resolution remote-sensing image segmentation. The DeeplabV3+ network is a deep neural network based on encoder-decoder architecture, which is commonly used to segment images with high precision. However, the segmentation accuracy of high-resolution remote-sensing images is poor, the number of network parameters is large, and the cost of training network is high. Therefore, this paper improves the DeeplabV3+ network. Firstly, MobileNetV2 network was used as the backbone feature-extraction network, and an attention-mechanism module was added after the feature-extraction module and the ASPP module to introduce focal loss balance. Our design has the following advantages: it enhances the ability of network to extract image features; it reduces network training costs; and it achieves better semantic segmentation accuracy. Experiments on high-resolution remote-sensing image datasets show that the mIou of the proposed method on WHDLD datasets is 64.76%, 4.24% higher than traditional DeeplabV3+ network mIou, and the mIou on CCF BDCI datasets is 64.58%. This is 5.35% higher than traditional DeeplabV3+ network mIou and outperforms traditional DeeplabV3+, U-NET, PSP-NET and MACU-net networks.

## 1. Introduction

In 1999, China successfully launched the first land resources remote-sensing satellite “Ziyuan-1”, with the launch of “Gao fen”, “Tian hui”, and “Zi yuan” series satellites, China’s remote-sensing satellite-mapping technology has made considerable progress since then. As remote-sensing technology advances, more and more useful information is extracted from remote-sensing images [1]. In addition to their high spatial resolution, high timeliness and large amount of information, high-resolution remote-sensing images provide a wealth of information [2,3]. High-resolution remote-sensing image feature classification plays an important role in surface-spatial information extraction, land resource management and natural resource protection. A semantic segmentation technology is used to classify the surface of high-resolution remote-sensing images. Each pixel of the image will be tagged with a category [4], and marked with the same colour on the targets of the category.

Traditional image segmentation techniques include threshold-based segmentation [5], clustering segmentation [6] and edge segmentation [7]. These methods need to manually select appropriate feature parameters, which have good results for segmenting simple scenes. However, in complex scenes, the segmentation effects of various targets vary greatly, and there are also many limitations. 

The rapid development of deep learning has enabled semantic segmentation to achieve pixel-level classification of images [8], so deep-learning networks are widely used in image segmentation, medicine, engineering technology and other fields. Zhao et al. [9] added a pyramid pooling module to the FCN [10] model and developed the PSP-Net network model to aggregate context information from different regions, thereby enhancing global information accessibility. Yang et al. [11] proposed skeleton monitoring based on MSB-FCN and showed that considerable progress has been made in applying semantic segmentation to medical images. Yu et al. [12] proposed a different crack detection method based on image by using deep convolutional neural network and Enhanced Chicken Swarm Algorithm (ECSA), and conducted comparative experiments with other crack detection methods on a variety of concrete crack samples. Fan et al. [13] proposed a microcrack detection method based on ResNet to detect microcracks on polycrystalline solar cells, which effectively improved the accuracy of detecting solar-cell defects. In 2021, on the basis of the U-Net [14] model, Li et al. [15] used asymmetric convolution to enhance the feature representation and feature extraction capability of the convolution layer, and effectively improved the problem of insufficient feature utilization of the U-Net model. The proposed MACU-Net network was tested on WHDLD and GID datasets, proving that the performance of the proposed network was better than that of the U-Net network. Chen et al. [16] used hole convolution on the basis of FCN to avoid the loss of pooled information, and then added the CRF (Conditional Random Fields) module at the output of the feature extraction network to propose the Deeplab network. Chen et al. [17] proposed the DeeplabV2 network. The network architecture introduced the ASPP (Atrous Spatial Pyramid Pooling) module, which improved the receptive field of the network without increasing the number of parameters, thus improving the network performance. The DeeplabV2 network takes conditional random fields and hole space pyramid pooling as the core. Later, Chen et al. [18] proposed the DeeplabV3 network. The network removes conditional random fields, uses hole-convolution cores for many times, and optimizes the ASPP module, so that the segmentation effect is better than the DeeplabV2 network. Since the features of the DeeplabV3 network do not contain too many shallow features, Chen et al. [19] proposed the DeeplabV3+ network. The DeeplabV3+ network is considered a new milestone in semantic segmentation technology. It proposes a classical coding and decoding structure based on the DeeplabV3+ network, which fuses shallow features with deep features, so as to improve the segmentation accuracy. Although the DeeplabV3+ network has shown leading advantages in multiple public datasets in the field of semantic segmentation, the segmentation of remote-sensing images is still not fine enough. 

In view of the fact that the DeeplabV3+ network needs to face the complex features of images in the process of image segmentation from remote sensing, this paper proposes to take the MobilenetV2 network as the feature extraction network, add an attention mechanism module in the network structure, and introduce focal loss to balance the loss. The segmentation performance of traditional DeeplabV3+, U-NET, PSP-NET and MACU net networks on remote-sensing datasets is compared. The experimental results show that the segmentation performance of this method is higher than that of other compared networks, and the training time of the network is significantly reduced. It shows that the improved method in this paper can effectively improve the network segmentation ability and reduce the network training cost.

## 2. Related Research 

### 2.1. Deeplabv3+ Network Model 

Since the DeeplabV3+network was proposed, it has often been used for high-precision image segmentation due to its excellent image segmentation ability [20,21]. The DeeplabV3+ network has been widely used in remote-sensing image segmentation in recent years [22,23,24]. The DeeplabV3+ network can mine context information on a multi-scale basis, and can also use the method of reconstructing image spatial information to determine the boundary of objects. For the input characteristic diagram X of ASPP module in the network structure, its output characteristic diagram (y) is: (1)$$\;y=\overset{K}{\underset{k=1}{\sum }}x\left[i+r\times k\right]w\left[k\right]$$
where *i* is the input signal, *w*[*k*] is the filter value, *R* is the expansion rate, and *K* is the hole convolution length.

Deeplabv3+ network introduces a large number of hole convolutions in the encoder module, which makes the network increase the receptive field without losing information. The receptive field calculation of hole convolution is consistent with the standard convolution kernel, and the value *k* of the hole convolution receptive field is: (2)K=k+(k−1) (r−1) 
where *k* is the size of the original convolution kernel; and R is the size of the actual convolution kernel of hole convolution.

The DeeplabV3+ network model structure mainly refers to the proposed coding and decoding structure, which not only improves the network segmentation effect, but also pays attention to the boundary information. The structure can be divided into encoder and decoder. The encoder is used for feature processing, which is divided into three steps: (1) DeeplabV3+ uses the modified aligned Xception feature extraction network for feature extraction on the network architecture, and finally generates two effective feature layers, namely, shallow features and deep features. Because the down-sampling of the deep characteristic layer is more than that of the shallow characteristic layer, the height and width of the shallow characteristic layer are larger than that of the deep characteristic layer. (2) The extracted deep features are input into the ASPP module, in which 1 × 1 convolution and expansion rates are 6, 12 and 18 respectively. 3 × 3 convolution and image global average pooling. The hole convolution with different expansion rates is used to improve the receptive field of the network, so that the network has different characteristic receptive conditions. (3) Stack the feature layers through 1 × 1 convolution to adjust the number of channels, and finally obtain the multi-scale feature of 256 channel digits. Shallow features generated by modified aligned Xception the generated shallow features enter the decoder, and the multi-scale features are up-sampled by 4 times, and then fused with the shallow features, after 3 × 3 convolution is used for feature extraction. Finally, the output picture is consistent with the input picture through 4 times of up-sampling, and the prediction result is obtained through Softmax function. Figure 1 shows the DeeplabV3+ network structure. 

### 2.2. Modified Aligned Concept Network Model 

The modified aligned Xception network is improved based on the Xception network, which is a network model proposed by the Google team in 2016. The Xception network structure consists of three flows, namely, entry flow, middle flow and exit flow [25]. The network has a total of 36 convolution layers. A deep separable convolution is introduced into the Xception network structure. Firstly, each channel is divided into a 3 × 3 convolution operation, and then a 1 × 1 convolution operation, and finally the results are merged. 

The residual structure is also introduced into the Xception network structure, which plays an important role in the model effect. Suppose the input is x and the expected output is y. In the traditional linear network structure, the learning objective f (x) = y; In the residual structure, the input x is directly transferred to the output as the initial result, and the learning objective f (x) = y − x, that is, the learning objective is changed from the original y to the difference between the output and the input. 

A modified aligned Xception network makes two improvements to the Xception network. One is to reduce the computational complexity by using deep separable convolution to replace the maximum pooling operation in all Xception networks. Second, in every 3 × 3 normalized and ReLU activation functions are used after the convolution operation. Figure 2 shows the modified aligned Xception network structure. 

## 3. Improved DeeplabV3+ Network Model 

In the traditional DeeplabV3+ network model, the modified aligned Xception network is selected for feature extraction. Due to the large number of layers of the network, the large amount of parameters, and the common convolution in the ASPP module after feature extraction, the complexity of the model is large, which increases the difficulty of network training, and the network training speed is slower and the convergence is slower. In order to improve the network segmentation ability and training efficiency, this paper makes three improvements on the network structure of high-resolution remote-sensing image ground-object segmentation based on DeeplabV3+: (1) the modified aligned Xception feature extraction network used for feature extraction is changed to a lighter MobileNetV2 network. (2) In the MobileNetV2 feature extraction network and after the ASPP module, the Ca (Coordinate attention) module [26] is added to improve the segmentation accuracy of the model. (3) Optimize the loss function and introduce focal loss equilibrium loss to reduce the impact on the accuracy of model feature classification due to the large difference in the proportion of feature categories. Figure 3 shows the improved DeeplabV3+ network structure.

### 3.1. Optimized Feature Extraction Module 

In the coding layer, the original network used for feature extraction is changed to a more lightweight MobileNetV2 network, which is a lightweight network model proposed by the Google team in 2018 [27]. Compared with the modified aligned Xception network, the MobileNetV2 network has fewer parameters, a simpler model structure, and faster network training speed. MobileNetV2 network uses linear bottleneck structure and reverse residual structure [28]. 

In the feature extraction operation, the neural network extracts the useful information of the target, which can be embedded in the low dimensional subspace. The traditional network structure is normalized by convolution containing the ReLU activation function, but using the ReLU activation function in the low dimensional space will lose more useful information. In the linear bottleneck structure, the ReLU activation function is changed to the linear function to reduce the loss of useful network information. 

The reverse residual structure of MobileNetV2 network application consists of three parts. Firstly, 1 × 1 convolution to increase the dimension of input features, and then use 3 × 3 depth separable convolution for feature extraction, and then 1 × 1 convolution for dimension reduction. The specific network structure is shown in Table 1. 

### 3.2. Add CA Module 

The attention mechanism in deep learning selects from a large set of information the information that is more critical for the task at hand. The performance of image segmentation tasks can be better improved by combining the attention mechanism with the fast convolution [29]. The common attention mechanisms include channel attention mechanism [30] and spatial attention mechanism [31]. This paper adds a CA module in the MobileNetV2 feature extraction network and after the ASPP module. The attention mechanism added to the encoder is equivalent to the feature extraction process to obtain the input attention representation. Adding an attention mechanism to the decoder input can improve the situation that the encoder output is a single long tensor and cannot store too much information. 

The CA module first uses two one-dimensional global pooling operations to aggregate the vertical and horizontal input features into two independent direction-aware feature maps. Then the two feature maps embedded with direction-specific information are encoded into two attention maps, which capture the long-distance dependence of the input feature map in the spatial direction. Therefore, the position information can be saved in the generated attention map. Finally, two attention graphs are applied to the input feature graph by multiplication to emphasize the representation of interest. 

Channel attention is globally encoded by global pooling, but it is difficult to retain location information, which is the key to obtain spatial structure in visual tasks. In order to enable the attention module to capture remote interaction in space by using accurate location information, the global pooling is decomposed, and paired one-dimensional feature coding is performed according to Formula (3).
(3)$$\;z_{c}=\frac{1}{H\times W}\overset{H}{\underset{i=1}{\sum }}\overset{W}{\underset{j=1}{\sum }}x_{c}\left(i,j\right)$$

With the given input *x*, each channel is encoded along the horizontal and vertical coordinates using a pooled core of size (*h*, 1) or (1, *w*). Therefore, the output with height *h* of channel C can be expressed as:(4)$$\;z_{c}^{h}\left(h\right)=\frac{1}{W}\underset{0\leq i<W}{\sum }x_{c}\left(j,w\right)$$

Similarly, the output with width W of channel C can be expressed as:(5)$$z_{c}^{w}\left(w\right)=\frac{1}{H}\underset{0\leq i<H}{\sum }x_{c}\left(j,w\right)\;$$

Through the above transformation, the features are aggregated along two directions, and two aggregation feature graphs are obtained. This section will stack the two generated aggregate characteristic graphs, and then use 1 × 1 convolution transform function f_ 1 change it:(6)$$f=\delta \left(F_{1}\left[z^{h},z^{w}\right]\right)\;$$
where [·, ·] represents the concatenation operation along the spatial dimension, δ is a nonlinear activation function, *f* is an intermediate feature map encoding spatial information in the horizontal and vertical directions, γ is the reduction ratio of the control module size. f is decomposed into two independent tensors $f^{h}$∈$R^{C/\gamma \times H}$, $f^{w}$∈$R^{C/\gamma \times w}$. Using the other two 1 × 1 convolution transforms $F_{h}$ and $F_{w}$ and converts $F_{h}$ and $F_{w}$ into tensors with the same number of channels to input X, and obtains:(7)$$g^{h}=\sigma \;\left(F_{h}\;\left(f^{h}\right)\right)$$
(8)$$g^{w}=\sigma \;\left(F_{w}\;\left(f^{w}\right)\right)$$

σ is the sigmoid activation function. In order to reduce the computational overhead and complexity of the model, an appropriate reduction ratio is used γ to reduce the number of channels of f. Then, the output $g^{h}$ and $g^{w}$ are extended and used as attention weights respectively. Finally, the output Y of the CA module can be expressed as:(9)$$y_{c}\left(i,j\right)=x_{c}\left(i,j\right)\times g_{c}^{h}\left(i\right)\times g_{c}^{w}\left(\;j\;\right)$$

### 3.3. Optimizing the Loss Function

For a deep-learning model, the neural network weights in the model are trained through loss back propagation [23]. Because the high-resolution remote-sensing image acquired has a large ground range and the proportion of different types of ground objects is different, the network training will be dominated by larger objects, and the classifier will classify other types of objects as larger objects, which will affect the performance of the segmentation network. Therefore, for the purpose of balancing the loss, this paper uses focal loss to deal with the problem of unbalanced proportion of classified objects. The calculation formula of focal loss is shown in Equation (10).
(10)$$FL=-a_{c}{\left(1-p_{c}\right)}^{\gamma }\log\;\left(p_{c}\right)$$
where, through weight $a_{c}$ to balance the uneven proportion of samples, $a_{c}$’s range is [0, 1], γ is a hyper parameter, γ’s range is [0, 5], When the value of γ is 0, focal loss is the traditional cross entropy loss, and $p_{c}$ is the prediction probability of different categories. The lower the value of $p_{c}$ the harder it is to classify, the higher the ${\left(1-p_{c}\right)}^{\gamma }$ is, and the smaller the value of ${\left(1-p_{c}\right)}^{\gamma }$ the easy it is to classify. For simple and easy samples, the larger the corresponding prediction probability, the smaller the corresponding weight. The smaller the prediction probability, the greater the weight of the complex samples.

## 4. Experiment and Result Analysis

### 4.1. Experimental Data

This paper uses the Wuhan dense annotation dataset (WHDLD) and CCF BDCI to test the effectiveness of the improved DeeplabV3+ network. WHDLD contains 4940 images taken in Wuhan by gaofen-1 and ziyuan-3 satellite sensors 256× 256 pixel RGB image with a spatial resolution of 2 m. The categories of image annotation in the dataset are divided into six categories: bare soil, buildings, sidewalks, water bodies, vegetation and roads. CCF BDCI is the dataset of the 2020 Baidu independent competition entitled “remote-sensing image parcel segmentation”, which contains 140,000 pieces of 256 × 256 pixel RGB image with a spatial resolution of 2 m. The categories of image annotation in the dataset are divided into seven categories: buildings, cultivated land, forest land, water, roads, grasslands and others.

### 4.2. Experimental Environment and Evaluation Criteria

The CPU used in this experiment is Intel i7-10700, the operating system is windows10, the GPU is NVIDIA Geforce RTX2060S, the video memory is 8 g, and the development environment is pytorch1.9.0 and python3.

In this study, the segmentation performance of high-resolution remote-sensing image datasets is evaluated. The average intersection union ratio (mIoU), average pixel accuracy (mPA) and average recall rate (mRecall) are selected as the indicators to evaluate the network segmentation performance. mIoU formula is shown in Equation (11), mPA formula is shown in Equation (12), and mRecall formula is shown in Equation (13).
(11)$$\mathrm{mIoU}=\frac{1}{n}\overset{n}{\underset{i=0}{\sum }}\frac{p_{ii}}{\sum_{j=0}^{n}p_{ij}+\sum_{j=0}^{n}p_{ji}-p_{ii}}$$
(12)$$\;\mathrm{mPA}=\frac{1}{n}\overset{n}{\underset{i=0}{\sum }}\frac{p_{ii}}{\sum_{i=0}^{n}\sum_{j=0}^{n}p_{ij}}$$
(13)$$\mathrm{mRecall}=\frac{1}{n}\overset{n}{\underset{i=0}{\sum }}\frac{p_{ii}}{p_{ii}+\sum_{i=0}^{n}p_{ji}}$$
where *n* is the total category, $p_{ij}$ represents the number of pixels that predict class *I* as class *J*, $p_{ij}$ represents the number of pixels that predict class *J* as class *I*, $p_{ii}$ represents the number of pixels that predict class *I* as class *I*.

### 4.3. Experimental Process

For the dataset WHDLD, training images were selected randomly from 60% of the images, 20% of the images as the verification set, and the remaining 20% of the images as the test set. A total of 14000 images were randomly selected for the dataset CCF BDCI, of which 60% were used as the training set, 20% as the verification set, and the remaining 20% as the test set. All models are implemented with Pytorch. The learning rate is 0.0005 and the training iteration number was 1000 times. In order to improve the accuracy of model segmentation, the MobileNetV2 pre-training weight is loaded before model training, and the focal loss function is used. When the control variables are the same, it is obtained through multiple experiments that the segmentation accuracy of the model is the highest when γ is 2,$\;a_{c}$ is 0.25.

### 4.4. Ablation Experiment

The effectiveness of changing the original network used for feature extraction into MobileNetV2 network, adding a CA module after the feature extraction module and ASPP module, and setting up four different schemes for experiments. The table below shows the experimental results.

Scheme 1 based on the traditional DeeplabV3+ network structure, the feature extraction network is replaced by MobileNetV2 network.

Using scheme 1 as a basis for scheme 2, add CA module to MobileNetV2 network;

Using scheme 1 as a basis for scheme 3, add CA module after ASPP module;

Using scheme 1 as a basis for scheme 4, enter the CA module after the ASPP module in the. MobileNetV2 network neutralize the ASPP module and then enter the CA module.

The initial learning rate is 0.0005 and the training iteration is 1000 steps. Table 2 shows the experimental results of different schemes on the WHDLD dataset, and Table 3 shows the experimental results of different schemes on the CCF BDCI dataset. Figure 4 and Figure 5 show the results of the ablation experiment. It can be seen that the network performance of scheme 4 exceeds that of other schemes. It shows that adding CA module in the feature extraction module and after ASPP module can effectively improve the accuracy of model segmentation, but adding CA module in the feature extraction module and after ASPP module can improve the effect better.

### 4.5. Comparison of Segmentation Performance of Different Methods

In order to further verify the segmentation performance of the improved DeeplabV3+ model proposed in this paper, this method is compared with the traditional WHDLD, U-Net, PSP-Net and MACU-Net networks.

Table 4 shows the experimental results of different methods on the WHDLD dataset, and Table 5 shows the experimental results of different schemes on the CCF BDCI dataset. It can be seen that the performance of the method proposed in this paper exceeds that of other networks. In the results of WHDLD dataset, the MIoU of the proposed method is 6.44% higher than that of u-net and 2.39% higher than that of MACU-Net. In the results of CCF BDCI dataset, the method proposed in this paper is 5.35% higher than the traditional DeeplabV3+ network MIoU. The results show that the traditional DeeplabV3+network has better segmentation performance than U-Net and PSP-Net in the task of high-resolution remote-sensing image segmentation. Also, it proves that the improved method proposed in this paper improves the segmentation performance of the network in high-resolution remote-sensing image segmentation.

Table 6 shows the comparison of the time spent by different methods to train 1000 steps on the WHDLD dataset and the number of parameter quantity, from which it can be seen that the training time and parameters of the model proposed in this paper are significantly reduced compared with the traditional DeeplabV3+network. Compared with MACU-Net, this method has less training time and more parameters. This is because the MACU-Net model has fewer parameters, but its model structure is more complex, which increases the model training time. In general, the MobileNetV2 network is proposed as the feature extraction network, and the CA module is added after the feature extraction module and ASPP module, which greatly reduces the amount of parameters and training time of the model.

Figure 6 shows the prediction results of the proposed method and the traditional DeeplabV3+network. On the left is the result of the WHDLD dataset, and on the right is the result of the CCF BDCI dataset. From this, it can be seen that the method in this paper is better than the traditional DeeplabV3+network segmentation. The main purpose of this method is to improve the segmentation effect of roads and buildings more. Because the categories of roads and buildings account for less in the dataset, the traditional DeeplabV3+network has poor segmentation accuracy in the categories that account for less. The results prove the necessity of adding CA module and introducing focal loss function.

### 4.6. Comparison with Existing Surface Feature Classification Methods of High-Resolution Remote-Sensing Images

At present, there are few research results on remote-sensing image ground feature classification based on depth neural network. The literature [32,33,34,35] proposed the research on high-resolution remote-sensing image ground feature classification based on support vector machine, ReliefF algorithm, genetic algorithm, tabu search algorithm, respectively, and an adaptive neuro-fuzzy inference system. The comparison between the ground feature classification method based on the depth neural network proposed in this paper and the existing ground feature classification methods is shown in Table 7.

Based on Table 7, the following improvement is observed compared with the feature classification method in reference [32,33,34,35]: 

(1)The ground feature classification methods adopted in references [32,33,34] need to manually select the spectrum, texture, geometry, shadow, background and geoscience auxiliary features of the ground feature categories of remote-sensing images, and the segmentation results depend on the advantages and disadvantages of the image feature selection; In this paper, the use of depth neural networks can automatically extract the features of images, which can give full play to the advantages of depth neural networks in image segmentation, and provide a new idea for the study of ground object classification of remote-sensing images.(2)The advantage of the method adopted in literature [35] is that only a small number of images are needed as the model training set, and the RGB value of each pixel in the image and an additional feature related to the local entropy information are extracted. The final classification accuracy is 0.06% different from that of the deep learning model (FCN) that requires a lot of image training. The classification rules of the method in literature [35] are explained in the paper. When the RGB value of a pixel value in the image of the WHDLD dataset has a high value, it is classified as a building class. The classification results also classify roads and sidewalks as building classes. The method proposed in literature [35] takes a long time to train on a small number of image datasets, and the classification rules of the model limit its inability to accurately classify labels with similar RGB values. In contrast, although the deep-learning method requires a large number of images as the training set, it can be more effectively segment the image dataset with more labels. (3)The attention mechanism is applied to the remote-sensing image ground object classification model based on the depth neural network, and the lightweight MobilenetV2 network is used as the backbone feature extraction network, which improves the ground-object classification accuracy of the model, results in a reduction of model parameters and a lower training cost.

## 5. Conclusions 

Based on the traditional DeeplabV3+ network, this paper uses the MobileNetV2 network as the backbone feature extraction network, and adds a CA module after the feature extraction module and the ASPP module. In addition, in view of the different proportion of different types of ground objects in the high-resolution remote-sensing image dataset, the focal loss equalization loss is introduced for high-resolution remote-sensing semantic segmentation. Experiments on WHDLD datasets and CCF BDCI datasets have fully confirmed that the proposed method outperforms the benchmark method.

The later work of this paper mainly includes four aspects: one is to improve the segmentation accuracy of the network, mainly aiming at the problem that the segmentation accuracy of the network is not high in road and building categories. Only by improving the categories with low segmentation accuracy can we effectively improve the segmentation accuracy of the network for high-resolution remote-sensing images. The second is to select more high-resolution remote-sensing images and more types of ground objects for further research and experiments. In addition to the network itself, there is also interference in the image that affects the network segmentation accuracy. So the third is to study how to reduce the interference of noise, shadow and other uncertain factors in the image to the network segmentation accuracy. The fourth is to continue to improve the network model and improve the segmentation accuracy of the network.

## Data Availability

I choose to exclude this statement.

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
