# Peer review of "Research on Ground Object Classification Method of High Resolution Remote-Sensing Images Based on Improved DeeplabV3+"

_sensors, 2022, doi:10.3390/s22197477_

Round 1

Reviewer 1 Report

In the experiments, I would suggest adding the results of transformer-based methods as they are now considered SOTA models in several domains. Please analyze also the results in terms of a number of parameters and inference time. It will be interesting to work also on well-known datasets for comparison purposes.

.

Author Response

Please find the answers attached.

Reviewer 2 Report

In this manuscript, the authors propose improvements over the DeepLabV3+ network for semantic segmentation in the context of remote sensing imagery. The contribution is incremental to existing work, but still improves results on two challenging datasets over several baselines. However, the quality of presentation of the article is rather poor. The manuscript should be proofread by a native speaker, as it contains many errors. There are also formatting problems. The resolution of the figures is poor and they contain Chinese characters. Finally, I suggest enriching the related work with other recent work on the same dataset, such as [http://ceur-ws.org/Vol-3074/paper15.pdf].

Reviewer 3 Report

This manuscript improved DeeplabV3+ model for ground object classification based on high resolution remote sensing images, where  MobilenetV2 network was used as the backbone feature extraction network, and attention mechanism module was added after feature extraction module and ASPP module to introduce Focal Loss balance. Finally, the experimental remote sensing dataset was used to validate the performance of the proposed model, with satisfactory results. The proposed model has the highest mIOU, which outperforms other models. Overall, the topic of this research is interesting, and the manuscript was well organised and written. The detailed comments are summarised as follows.

1.       The contribution and innovation of the manuscript should be clarified clearly in abstract and introduction.

2.       Broaden and update the literature review in engineering applications of CNN or deep learning, such as image processing, segmentation, structural defect recognition. E.g. Crack detection of concrete structures using deep convolutional neural networks optimized by enhanced chicken swarm algorithm. 

3.       How did the authors set the hyperparameters of the proposed model for optimal classification performance?

4.       How about the robustness of the proposed model against noise or other uncertainties?

5.       A parametric study is necessary to analyse the model architecture.

6.       More future research should be included in the conclusion part.

Round 2

Reviewer 1 Report

The authors have answered my comments.